# Reliability-Based Serviceability Limit State Design of a Jacket Substructure for an Offshore Wind Turbine

**Jianhua Zhang [1], Won-Hee Kang [2], Ke Sun [3,*] and Fushun Liu [4]**

[1]  College of Aerospace and Civil Engineering, Harbin Engineering University, Harbin 150001, China
[2]  Centre for Infrastructure Engineering, Western Sydney University, Sydney, NSW 2751, Australia
[3]  College of Shipbuilding Engineering, Harbin Engineering University, Harbin 150001, China
[4]  College of Engineering, Ocean University of China, Qingdao 266100, China
*  Correspondence: sunke@hrbeu.edu.cn

**Abstract:** The development of a structurally optimized foundation design has become one of the main research objectives for offshore wind turbines (OWTs). The design process should be carried out in a probabilistic way due to the uncertainties involved, such as using parametric uncertainties regarding material and geometric properties, and model uncertainties in resistance prediction models and regarding environmental loads. Traditional simple deterministic checking procedures do not guarantee an optimized design because the associated uncertainties are not fully considered. In this paper, a reliability analysis framework is proposed to support the optimized design of jacket foundations for OWTs. The reliability analysis mainly considers the serviceability limit state of the structure according to the requirements of the code. The framework consists of two parts: (i) an important parameter identification procedure based on statistical correlation analysis and (ii) a finite element-simulation-based reliability estimation procedure. The procedure is demonstrated through a jacket structure design of a 3 MW OWT. The analysis results show that the statistical correlation analysis can help to identify the parameters necessary for the overall structural performance. The Latin hypercube sampling and the Monte Carlo simulation using FE models effectively and efficiently evaluate the reliability of the structure while not relying on a surrogate limit state function. A comparison between the proposed framework and the deterministic design shows that the framework can help to achieve a better result closer to the target reliability level.

**Keywords:** offshore wind turbine; jacket structure; serviceability limit state; statistical correlation analysis; reliability analysis

## 1. Introduction

Renewable offshore wind energy is regarded as one of the leading alternative options to reduce greenhouse gas emission and to promote independence from fossil fuels. Its capacity has rapidly expanded worldwide in recent years. The Global Wind Energy Council [1] reported that the global cumulative capacity of offshore wind energy amounted to 23,140 MW in 2018. There was a 4496 MW increase between 2017 and 2018, and this was the most significant yearly addition to the capacity up until 2018. Further to the rapid development of offshore wind energy until now, the global demand is still growing for wind energy production [2,3]. The total installed capacity of offshore wind power is expected to be increased to 120 GW by 2030.

A jacket foundation is a fixed supporting structure of an offshore wind turbine (OWT). It transfers all loads from the OWT to the ground through an allowable deflection. The permissible deviation formulates the serviceability limit state (SLS) of an OWT in its structural design to ensure the safe operation and any visually exposed concerns. For this, the tilt at the hub lever should be strictly

considered and controlled according to engineering standards such as DNV [4] and API [5]. If the angle exceeds the allowable threshold, the operation of the OWT needs to be stopped. Also, the construction of the foundation makes up approximately 20–30% of the total cost [6,7]. Therefore, the design of the foundation in a fully optimized way has become one of the leading research objectives in the field of offshore wind energy.

Decision-making in the design of an OWT foundation needs to consider many sources of uncertainties, such as material and geometric uncertainties, and the model uncertainties in resistance prediction models and in the environmental loads. These uncertainties are often not rigorously or thoroughly considered in the design phase, and approximations are usually made in deterministic or semi-probabilistic structural design formats. Therefore, the achieved design is usually not fully optimized and probabilistic analysis on the structural response is needed to meet the target reliability level.

Some studies were conducted to develop probabilistic approaches for the structural design of OWT foundations. Barbato et al. [8] discussed the effects of uncertainties embedded in structural parameters to the response of a jacket OWT. Mardfekri [9] proposed probabilistic models for shear, moment, and deformation on the mono-pile foundation of offshore wind turbines to generate the fragilities according to the serviceability and ultimate limit states. Liu et al. [10] proposed a new frequency-domain response estimation method for floating structures by dealing with fluid memory effects from the viewpoint of signal decomposition. El-Din and Kim [11] studied the sensitivity of the seismic structural response of a jacket platform concerning uncertain modeling variables using the tornado diagram and the first-order-second-moment (FOSM) method. Yang et al. [12] proposed a reliability-based design optimization (RBDO) method for a tripod-type OWT considering the requirements for dynamic responses to decrease the weight and cost of the foundation. Liu et al. [13] proposed an iterative noise extraction and elimination method that aims at solving the difficulty of modal parameter identification caused by contaminated high-energy components in measured signals. Vahdatirad et al. [14] found the uncertainties regarding soil properties and proposed an asymptotic sampling method to estimate the probabilistic distribution of stiffness for the OWT on a monopile foundation. However, the uncertainties in the response prediction models and the structural parameters together were not considered in their study. Sergio et al. [15] used a reliability-based approach and a probabilistic model to establish the design and limit state equations for the fatigue failure of a structure. Ziegler [16] developed a design clustering method for mono-pile OWTs by estimating probabilistic fatigue load containing significant uncertainties that are commonly addressed in the safety factors in design standards. Liu et al. [17] proposed a novel frequency-domain transient response estimation method to obtain reliable estimations of the dynamic responses of high-rising marine structures, such as offshore wind turbines, with obvious nonzero initial conditions.

To bridge the gap between these probabilistic approaches and the original structural design of OWT foundations, this study proposes a reliability analysis framework consisting of an identification procedure for important parameters based on statistical correlation analysis and a reliability estimation procedure based on FE-simulation. The proposed framework aims to achieve the reliability-based optimized structural design of jacket foundation for OWTs.

## 2. Proposed Reliability Analysis and Design Framework

This section presents the proposed reliability analysis and design framework consisting of an important parameter identification procedure, reliability analysis, and reliability-based design. A statistical correlation analysis is used to identify the importance of each of the uncertain model parameters in affecting overall structural performance.

### 2.1. Correaltion-Based Important Parameter Identification and Reliability Analysis

Probabilistic approaches can incorporate the uncertainties and variations of input parameters into a prediction model such as a finite element model [18]. For an effective probabilistic analysis, it is important to reduce the dimension of the parameters for sampling by identifying some important parameters. The approach based on statistical correlation analysis is provided below to identify such important parameters.

In a probabilistic analysis, it is assumed that the input parameters' statistical distributions are given. The statistical distributions can be defined by mathematical functions, such as normal/Gaussian and lognormal distributions. Then, by adopting the concept of the Monte Carlo simulation (MCS), the statistical distribution of the output parameters, i.e., structural responses, can be obtained by running the model many times according to the input parameters' random distributions. During this process, a vast array of randomly generated input parameter values and the calculated output values show parameter sensitivities and the statistical relationship between the design parameters (input parameters) and the structural performance (output parameters).

To further describe this procedure, we first express a structural response of the target jacket as follows:

$$Y = g(x_1, x_2, x_3 \ldots x_i \ldots x_n) \tag{1}$$

where $Y$ denotes the response of a structure, such as strain, stress, deformation, and natural frequency; $x_i$ is one of the possible uncertainty considerations in the paper; and $g(\cdot)$ is a function that represents the prediction of the structural responses estimated using the finite element model in this study. In this study, the Spearman rank-order correlation coefficient (SRC) is adopted to represent the statistical correlation between the structural response and the design input variables. The SRC is calculated in the following way:

$$SRC = \frac{\sum\limits_{i}^{n} (R_i - \overline{R})(S_i - \overline{S})}{\sqrt{\sum\limits_{i}^{n} (R_i - \overline{R})^2} \sqrt{\sum\limits_{i}^{n} (S_i - \overline{S})^2}} \tag{2}$$

where $R_i$ and $S_i$ are the random representations of $x_i$ and $Y$, respectively, for n samples. $\overline{R}$ and $\overline{S}$ are the average values of $R_i$ and $S_i$, respectively. *SRC* represents the degree of a linear statistical relationship between the two variables, $R$ and $S$. The larger the absolute value of *SRC*, the stronger the degree of the linear relationship between the input and output values. A positive value suggests that output is positively related to the input, while a negative value of *SRC* indicates the output is inversely related to the input. The SRC value lies within $-1$ and 1. Using *SRC*, the parameters with the *SRC* values closer to 1 or $-1$ are selected as important parameters. For the evaluation of *SRC*, MCS with Latin hypercube sampling (LHS) is used due to its computational efficiency in random samplings and straightforward implementation. In MCS analysis, geometric parameters, such as the diameters and thickness of structural components, and material property parameters are considered as random input/design parameters, and their statistical distributions are taken from the literature, as seen in Table 1. The maximum displacement at the hub level $D_{max}$ and the frequencies are taken as output parameters or structural responses. The statistical correlation analysis is repeatedly conducted between each of the random design parameters and the structural response using *SRC*.

**Table 1.** Random input parameters specifications.

| Random Input Parameters | Symbol | Mean | *c.o.v.* | Distribution Type | References |
|---|---|---|---|---|---|
| Elastic modulus (GPa) | TM | 210 | 7.6% | Gaussian | [19] |
| Yield strength (MPa) | YS | 355 | 6.8% | Lognormal | |
| Outer diameter of central column (mm) | D1 | 4740 | 10% | Gaussian | |
| Thickness of central column (mm) | T1 | 60 | 10% | Gaussian | |
| Outer diameter of top brace bottom (mm) | D2 | 1400 | 10% | Gaussian | |
| Thickness of top brace bottom (mm) | T2 | 60 | 10% | Gaussian | |
| Outer diameter of top brace top (mm) | D3 | 2000 | 10% | Gaussian | |
| Thickness of top brace top(mm) | T3 | 60 | 10% | Gaussian | |
| Outer diameter of leg (mm) | D4 | 1600 | 10% | Gaussian | |
| Thickness of leg (mm) | T4 | 30 | 10% | Gaussian | |
| Outer diameter of X-brace in top (mm) | D5 | 760 | 10% | Gaussian | |
| Thickness of X-brace in top (mm) | T5 | 28 | 10% | Gaussian | |
| Outer diameter of X-brace in bottom (mm) | D6 | 760 | 10% | Gaussian | |
| Thickness of X-brace in bottom(mm) | T6 | 28 | 10% | Gaussian | [20,21] |
| Outer diameter of anchorage pile (mm) | D7 | 2100 | 10% | Gaussian | |
| Thickness of anchorage pile (mm) | T7 | 50 | 10% | Gaussian | |
| Outer diameter of pile sleeve (mm) | D8 | 1600 | 10% | Gaussian | |
| Thickness of pile sleeve (mm) | T8 | 40 | 10% | Gaussian | |
| Web height of the hoop beam (mm) | HW | 1200 | 10% | Gaussian | |
| Web thickness of the hoop beam (mm) | TW | 25 | 10% | Gaussian | |
| Flange width of the hoop beam (mm) | WF | 500 | 10% | Gaussian | |
| Flange thickness of the hoop beam (mm) | TF | 30 | 10% | Gaussian | |
| Outer diameter of tower top (mm) | TD1 | 2800 | 10% | Gaussian | |
| Thickness of tower top (mm) | TT1 | 60 | 10% | Gaussian | |
| Outer diameter of tower bottom (mm) | TD2 | 4700 | 10% | Gaussian | |
| Thickness of tower bottom (mm) | TT2 | 30 | 10% | Gaussian | |

After identifying important parameters, a full probabilistic analysis can be effectively performed by considering the uncertainties of these parameters. Various approaches are available to perform a full probabilistic analysis including the first-order reliability method (FORM) [22], the second-order reliability method (SORM) [23], the MCS method [24], the response surface method [25], and their advanced forms. To estimate the reliability of the structure against the SLS, this study uses MCS together with LHS. LHS is a stratified sampling technique used to represent all possible outcomes of random variables in the simulation [26]. This method is chosen in this study because it does not rely on an approximation on the limit state function or the reliability estimation, and the SLS reliability calculation does not require many FEA calls compared to the ULS reliability calculation.

The reliability index $\beta$ is defined using the following relation with the failure probability $P_f$:

$$\beta = -\Phi^{-1}(P_f) \tag{3}$$

where $\Phi^{-1}(\cdot)$ is the inverse standard normal cumulative distribution function [26]. In this formulation, the probability of failure $P_f$ can be estimated using MCS or non-simulation-based methods, such as FORM and SORM, but these non-simulation based methods have limitations in dealing with non-linear functions such as the limit state functions defined by FEA [27]. In MCS, the failure probability is calculated as follows:

$$P_f = \frac{N_f}{N} \tag{4}$$

where $N_f$ is the number of FEA simulations within the failure domain, and $N$ is the total number of FEA simulations in both the failure and safe domains. The precision of $P_f$ increases as $N$ increases, which can be represented by the coefficient of variation (*c.o.v.*) of the failure probability as follows:

$$c.o.v._{P_f} \simeq \sqrt{\frac{1 - P_f}{N \cdot P_f}} \tag{5}$$

In this equation, the *c.o.v.* of $P_f$ decreases as $N$ increases.

## 2.2. The Design Framework

In this study, a reliability-based framework is proposed to design a jacket OWT structure by combining the statistical correlation analysis procedure and the reliability analysis procedure described in previous sections. This framework aims to achieve a better cost-serviceability balance by consistently meeting the target reliability level compared to the deterministic design. A flowchart for this framework is presented in Figure 1.

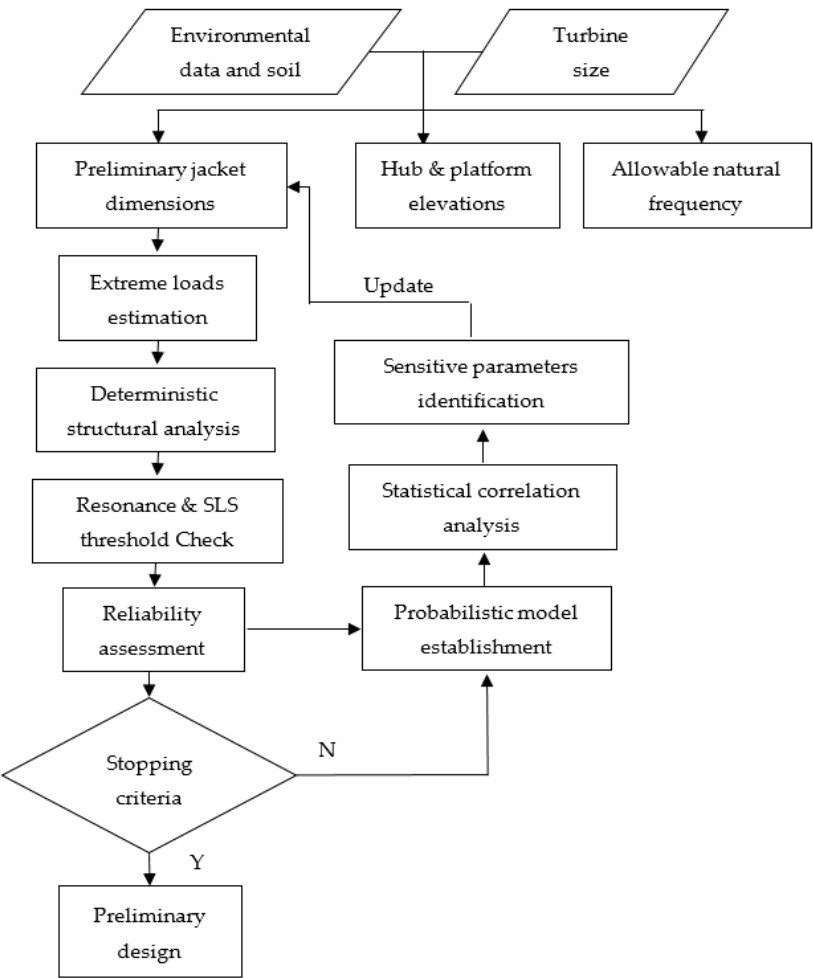

**Figure 1.** A flow chart for the proposed probabilistic preliminary design framework.

An iterative procedure is adopted in the design. First, in order to determine the platform and hub elevations and the initial jacket dimensions, an array of data is required, including environmental parameters such as wind, wave, and current; soil parameters such as the shear strength, submerged unit weight, and the strain; and turbine size related parameters such as power, blade diameter, and frequency range. The loads are estimated based on the initial dimensions of an OWT and the jacket structure dimensions. Then, structural responses including deformation and natural frequency are evaluated using FEA to check if they meet the SLS requirements in the codes and operation practice. Afterward, a further check is carried out by conducting a reliability analysis, in which MCS is used to check if the structural response meets the target reliability level against the SLS threshold values. In the reliability analysis, the uncertainties of the parameters and the structural model need to be included. If the evaluated reliability does not meet the target reliability, the design needs to be revised by changing the values of the critical parameters identified in the statistical correlation analysis until

the design requirement is met and the final preliminary design is obtained. It is noteworthy that if the support structure does not meet the strength requirements in subsequent ULS and FLS checks, the procedure will go back to the initial step.

## 3. The Computational Models of the Support Structures

The above correlation analysis and structural reliability estimation were applied to a target jacket substructure, and this process is described in this section. This section also provides the details of the finite element (FE) modeling and applied loads to the target structure.

### 3.1. Jacket Substructure Geometries

Figure 2 shows a sketch of a jacket structure for a 3 MW OWT. The jacket structure was analyzed at a water depth of 20 m. It consisted of a central column, four pile legs, four top braces, a hoop beam, and two layered X-braces. An inverted circular truncated cone structure with a diameter of 1.4 m–2 m was located at the tower and jacket connection. The piles were driven into the seabed to anchor the jacket substructure connected by the pile sleeve. The top and bottom of the jacket substructure were located 10 m and −20 m above the mean sea level (MSL), which included the pipe sleeve with a length of 4 m measured from the bracing bottom (BB) to the midline (ML). The bottom base of the jacket had an area of 15 m × 15 m, and the top base had an area of 9 m × 9 m. The upper conical tower mounted on the jacket is 65 m high, and the tower had diameters of 4740 mm at the base and 2860 mm at the top. The hub elevation was 75 m above MSL.

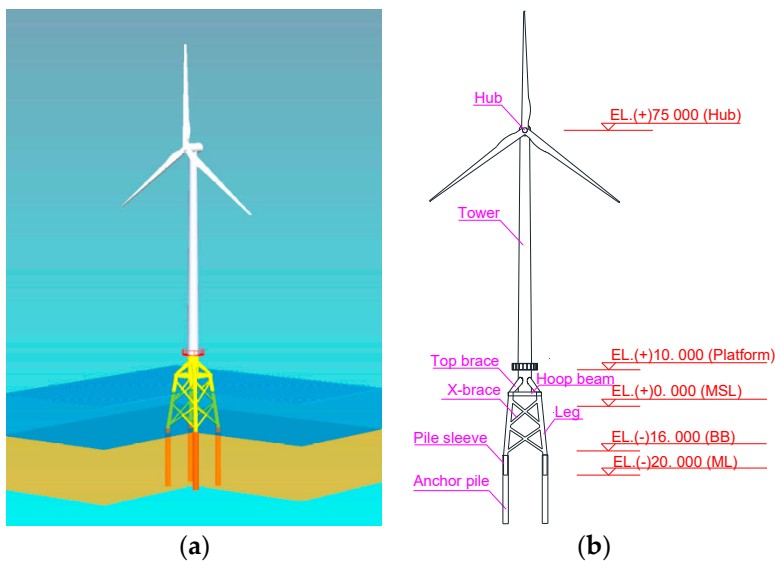

(**a**)　　　　　　　　　　　　　　　　　(**b**)

**Figure 2.** A sketch map of the jacket substructure: (**a**) three-dimensional view; (**b**) section.

### 3.2. Modeling for Finite Element Analysis

A 3D finite element (FE) model of the jacket is constructed using a commercial nonlinear FE code ANSYS by scripting in APDL, as shown in Figure 3. Beam 188 element was chosen for modeling the tower, piles, legs, and braces; and mass element was selected for the nacelle, rotor, and platform. The pile–soil interaction was modeled by firstly defining the lateral soil stiffness of the model. Generally, between the soil resistance (p) and its deformation (y), a nonlinear material constitutive relationship is established, i.e., the p–y curve described in the code of American Petroleum Institute (API) [5], which was used in this study. The p–y curve had a nonlinear form and was dependent on parameters such as depth, soil shearing stress, and the properties of soil. The nonlinear relationship in this curve was defined in the combine 39 element using the force–deformation (F-D) connection, in which F was the total force applied along the pile length.

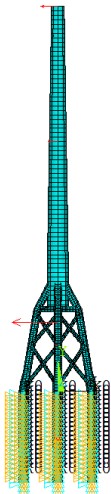

**Figure 3.** An FE model of the jacket.

The parameter details were established using the soil investigation data provided in a geotechnical report in Guishan Island, China [28]. As an illustration, Figure 4 shows the selected lateral stiffness of soil. In the material model, Young's modulus, yield strength, and the Poisson's ratio of steel were taken as 210 GPa, 355 MPa, and 0.3, respectively. The selected soil and steel material parameters are summarized in Tables 2 and 3.

**Table 2.** Soil material properties used in the FE model.

| Soil Layer | Soil Type | Soil Depth | | Effective Gravity (kN/m$^3$) | Design Shear Strength (kPa) | E$_{50}$ | ks |
|---|---|---|---|---|---|---|---|
| | | Top of the Soil Layer (m) | Bottom of the Soil Layer (m) | | | | |
| 1 | Very soft—hard silty clay | 0.0 | | 7.4 | 4 | 0.02 | |
| | | | 3.8 | 7.4 | 9 | 0.02 | |
| | | 3.8 | | 7.8 | 14 | 0.02 | |
| | | | 7.3 | 7.8 | 22 | 0.02 | |
| 2 | Medium dense silt | 7.3 | | 8.2 | $\varphi = 20°$ | | 5430 |
| | | | 8.1 | 8.2 | | | 5430 |
| 3 | Medium dense silt—fine silt | 8.1 | | 8.8 | $\varphi = 25°$ | | 5430 |
| | | | 13.3 | 8.8 | | | 5430 |
| 4 | Medium dense silt | 13.3 | | 9.0 | $\varphi = 20°$ | | 5430 |
| | | | 14.8 | 9.0 | | | 5430 |
| 5 | Hard silty clay | 14.8 | | 9.0 | 30 | 0.01 | |
| | | | 17.7 | 9.0 | | 0.01 | |
| 6 | Dense silt | 17.7 | | 9.2 | $\varphi = 25°$ | | 5430 |
| | | | 19.7 | 9.2 | | | 5430 |
| 7 | Dense silt | 19.7 | | 9.2 | $\varphi = 30°$ | | 10860 |
| | | | 26.7 | 9.2 | | | 10860 |
| 8 | Dense—very dense silt | 26.7 | | 9.2 | $\varphi = 25°$ | | 5430 |
| | | | 29.2 | 9.2 | | | 5430 |
| 9 | Hard—very hard silty clay | 29.2 | | 9.3 | 100 | 0.005 | |
| | | | 33.2 | 9.3 | 100 | 0.005 | |

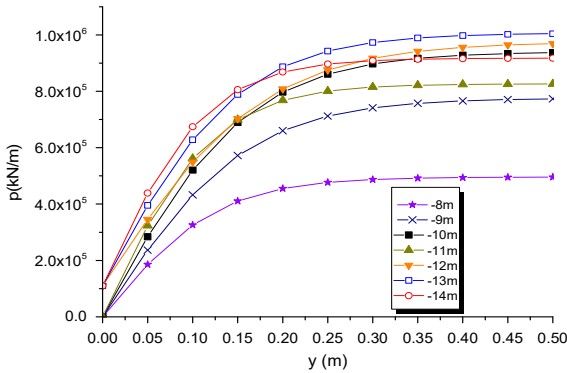

**Figure 4.** The p–y curve for the soil between −8 m and −14 m.

**Table 3.** Steel material properties used in the FE model.

| Young's Modulus (Steel) (GPa) | Yield Strength (Steel) (MPa) | Poisson's Ratio (Steel) |
| --- | --- | --- |
| 210 | 355 | 0.3 |

### 3.3. Load Models

OWTs and their support structures are often exposed to the harsh marine environment, and the environmental loads need to be rigorously considered in the limit states such as the ultimate limit state (ULS) and serviceability limit state (SLS). The essential information regarding the selection of the specific loads is described in standards and regulations such as DNV [4] and IEC [29]. In this study, primary loads from wind, wave, and gravity were considered, and appropriate load simplifications were made. The distributed loading, including wind loading and wave loading, was simplified as point loading to deal with the large-scale structural configurations and improve the computational efficiency. In addition, the current loading, which had a minor effect on the jacket foundation compared to wave loading and wind loading, was not considered in this study.

The steady and unsteady aerodynamic performance of the horizontal axis wind turbine was calculated via the blade element momentum theory, dynamic stall, and dynamic wake models. The aerodynamic properties of blades were obtained from the literature [30].

The drag load of the tower was calculated independently using the following equation from the API [5]:

$$F = \frac{1}{2} C_s \rho u^2(z) A \tag{6}$$

where $C_s$ is the shape coefficient, which is 0.5 in this study for cylindrical sections; $\rho$ is the density of air, which is 1.225 kg/m$^3$; and $A$ is the projected area of the tower facing the incoming wind. $u(z)$ is the wind speed (in meters per second) at height $z$ (in meters), and it can be calculated as follows:

$$u(z) = \left(\frac{z}{z_{ref}}\right)^m u(z_{ref}) \tag{7}$$

where $u(z_{ref})$ is the wind speed at the reference height z, and the exponential term $m$ is an empirical coefficient varying according to the atmosphere stability and is taken to be 0.143 in this study.

Diffraction effects were negligible for the waves as the jacket piles are slender cylinders, and the pile diameter $D$ is small compared to the wavelength λ. Therefore, the Morison formula can be used to calculate the wave force [4,31]. The horizontal force applied to the element of a cylinder at level $z$ is represented as follows:

$$F_W = \int dF \tag{8}$$

$$dF = dF_m + dF_d = C_m \, \rho \pi \frac{D^2}{4} \, \dot{u}_w dz + \ C_d \, \rho \frac{D}{2} |u_w| u_w dz \tag{9}$$

$$u_w = \frac{\pi H}{T} \frac{\cosh kz}{\sinh kd} \cos(kx - wt) \tag{10}$$

$$\omega = \frac{2\pi}{T} \tag{11}$$

$$k = \frac{2\pi}{L} \tag{12}$$

where $dF_m$ is the inertia force, and $dF_d$ is the drag force. $C_m$ and $C_d$ represent the inertia and quadratic drag coefficients, respectively. $\rho$ is the water density, and $D$ is a structural member's diameter. $\dot{u}_w$ and $u_w$ represent the acceleration and velocity of water in the horizontal direction, respectively. L is the wavelength. The positive direction of force is defined as the direction of wave propagation. The resulting force was calculated through the integration of the force over the length of the structure that is defined from the seabed to MSL. In this study, $C_m$ = 1.0 and $C_d$ = 2.0. In addition, the significant wave height H and the average wave period T were taken to be 14 m and 16.7 s, respectively, from the meteorological data at a specific site (about 5.3 km far from Guishan island in Zhuhai, Guangdong [28]). The water depth d was 15 m. The gravitation acceleration was set to be 9.81 m/s$^2$. The nacelle and rotor were simplified as a mass point at the top of the FE model, and the platform complied with the same simplification method but was deployed at the center of the platform. The turbines were also represented by a mass point at the top of the FE model but with a horizontal eccentric configuration of 1.5 m from the top of tower.

## 4. Heuristic Design of Jacket Substructure

In the preliminary design phase, the jacket substructure was first designed using the conventional trial-and-error design methodology, and the first draft dimensions of structural components of the jacket were determined based on prior experience. The draft solutions were then updated until the final design met the requirements in the codes of practice, such as DNV [4] and API [5]. After seven iterations, the details of the dimensions were determined, which are shown in Table 4.

**Table 4.** Component geometries of the support structure.

| Components | Size (Diameter) | Size (Thickness) | Unit |
|:---:|:---:|:---:|:---:|
| Anchor pile | 2000 | 50 | mm |
| Pile sleeve | 1600 | 40 | mm |
| Leg | 1600 | 30 | mm |
| X- brace | 760 | 28 | mm |
| Top brace | 1400 to 2000 | 60 | mm |
| Central column | 4740 | 60 | mm |
| Tower | 2800 to 4740 | 30 to 50 | mm |
| Nacelle and rotor | 163.3 | | ton |

To check the occurrence of resonance, modal analysis was first carried out to estimate the natural frequencies of the structure. Figure 5 shows the three modes shapes of the tripod structure. The deformation and natural frequency were estimated using ANSYS.

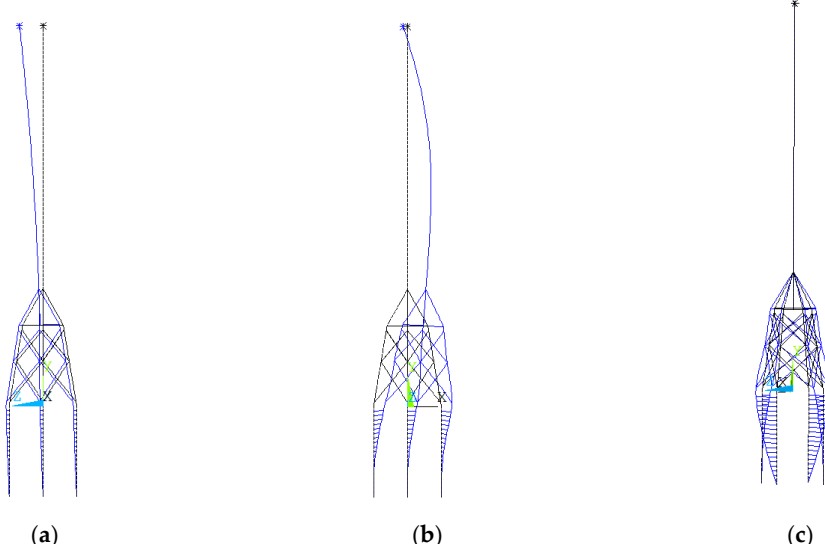

**Figure 5.** The first three mode shapes of the tripod structure: (**a**) frequency = 0.341 Hz, (**b**) frequency = 1.012 Hz, and (**c**) frequency = 3.212 Hz.

In this study, a three-blade V90-3 MW (Vestas, Aarhus, Denmark) wind turbine was adopted, and it had an operational rotational speed varying between 8.6 rpm and 18.4 rpm. According to the DNV code [4], resonance was avoided by ensuring that the first natural frequency was not within 10% of the rotor frequency (1P) range and within 10% of the corresponding 3P rotor harmonic range. For this turbine, the 1P frequency lay within the range between 0.143 Hz and 0.306 Hz, and the 3P frequency lay within the range between 0.429 Hz and 0.920 Hz. Figure 5 shows that the first natural frequency of the jacket was not within 10% of the 1P and 3P ranges, and resonance was thus avoided for the structure.

The contribution of the support structure was to transfer the applied loads to the ground bearing through an allowable deformation, which ensured the safe operation of the turbine. Regarding this allowable deformation, the DNV code [4] set a limit of 0.25 degrees in the tilt of the tower axis regarding the SLS criteria. Figure 6 represents the total rotation vector of the support structure. The maximum rotation at the hub level was $9.769 \times 10^{-3}$ rad corresponding to a tilt of 0.56 degrees, which exceeded the allowable tilt of 0.25 degrees.

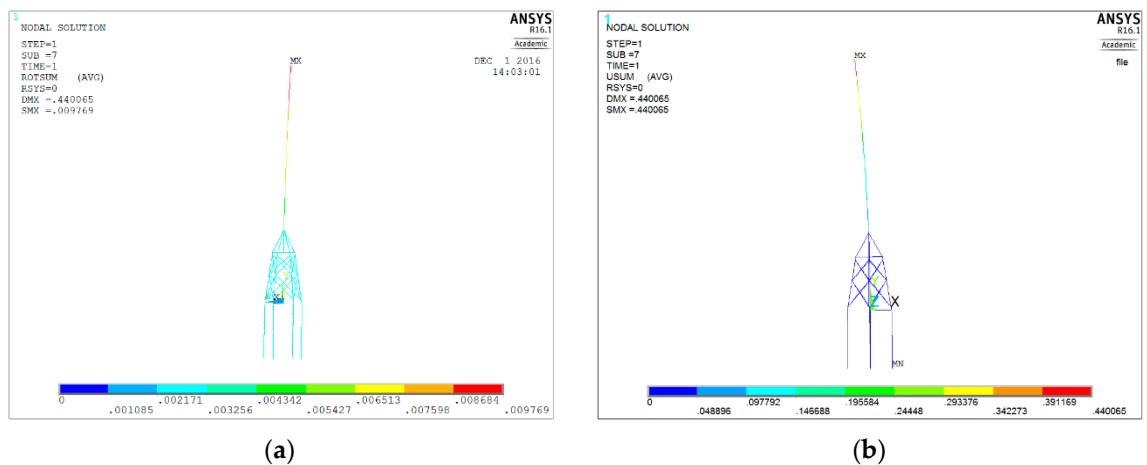

**Figure 6.** Deformation of the support structure: (**a**) total rotation vector; (**b**) total displacement vector.

In summary, the natural frequency of the support structure in the preliminary design was outside of the excitation frequencies, which avoided resonance. However, the deformation at the hub level

exceeded the required threshold value. Therefore, the design should be improved by increasing the stiffness of the support structure to meet the deformation requirement. This can be effectively achieved by identifying the key parameters that affect the performance of the structure the most.

There were more than 10 candidate design parameters that could be updated in the stepwise trial-and-error execution for the preliminary design. The primary concern for the designer is how to effectively identify the parameters that should be adjusted to meet the requirements in codes. The sensitivity analysis proposed in this study for identifying essential parameters and the reliability-based design procedure introduced in the next sections is necessary for this process.

## 5. Probabilistic Analysis

### 5.1. Important Engineering Demand Parameters Identification

To identify the important design parameters, the statistical correlation analysis mentioned in Section 2 for the target jacket substructure was carried out. Statistical properties of uncertain parameters associated with the structural model were summarized in Table 1. The geometric parameters were conservatively assumed such that the tolerance values were the same as the standard deviations of the parameters rather than directly adopting the variabilities found in practice.

The statistical correlation analysis results using *SRC* are shown in Figure 7, where the *SRC* between the maximum lateral deflection and each of the parameters for the structural geometry and material properties are provided. It is seen that the diameter of the tower bottom (TD2) had the most significant effect on the lateral deflection of the hub, showing an *SRC* value of −0.64. The negative value means that the increase in the diameter of the tower bottom resulted in the decrease in the displacement. The other parameters that had a significant effect include the elastic modulus (EM), the diameter of the top brace (D3), and the thickness of the tower bottom (TT2). The other parameters such as D1, D6, T3, D7, T4, D4, TT1, T6, and T7 had a relatively small influence on the maximum deflection, and D5, T5, TW, WF, and HW had almost of no significant effect on the maximum deflection.

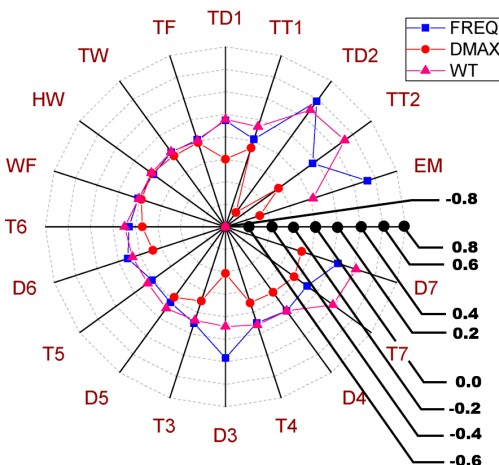

**Figure 7.** The *SRC*s between the design variables and the maximum deflection (DMAX), total weight (WT), and frequency (FREQ).

To consider the effect of each design parameter on the total cost, the *SRC* between each parameter and the total weight was considered. Weight was chosen as a measure of cost because it is related to the amount, transportation, and installation of construction material. Figure 7 shows that the design variable with the greatest effect on the total weight was the thickness of the tower bottom (TT2), and the variable with the second-greatest effect was the thickness of the outer diameter of tower bottom (TD2). Variables TD1, TT1, D4, T4, D5, T6, D3, T3, D6, T5, and hoop-beam-related parameters, i.e., TW, HW, WF, and TF, had negligible contributions to the total weight.

Compared to traditional offshore structures, OWT foundations impose a stricter requirement regarding the natural frequency to avoid resonance. Therefore, it is crucial for designers to know the correlation between the natural frequency and the design parameters. Figure 7 shows that the natural frequency of the jacket was highly dependent on the thickness of the tower bottom (TD2) and the elastic modulus (EM). In other words, adjustment of the thickness of the tower bottom is recommended if the natural frequency of the foundation structure does not meet the design requirements. On the other hand, T3, T4, T6, T7, D5, T4, T5, and the hoop-beam-related parameters, i.e., TW, HW, WF, and TF, showed no significant effect on the first-order natural frequency, and they did not significantly affect the design.

As seen from Figure 7, the statistical correlation coefficient was effective in identifying the important parameters instead of conducting parametric sensitivity analysis for each parameter. Also, by considering the identified parameters only, the size of the design optimization problem can be significantly reduced. In this study, out of 19 parameters, only 5 parameters with significant contribution to the deflection have been identified, and contributions from all other parameters have been identified as negligible. The five most important parameters and the five least important parameters are summarized in Table 5 with the ranks of 1–5 and 15–19.

**Table 5.** The ranks of the sensitive design parameters for the maximum deflection, the total weight, and the natural frequency.

| Rank | Maximum Deflection | First Natural Frequency | Total Weight |
|------|--------------------|-------------------------|--------------|
| 1 | TD2 | TD2 | TT2 |
| 2 | EM | EM | TD2 |
| 3 | D3 | D3 | D7 |
| 4 | TT2 | D7 | T7 |
| 5 | TD1 | TT2 | TD1 |
| 15 | TW | TT1 | TW |
| 16 | WF | WF | HW |
| 17 | TF | TF | WF |
| 18 | T5 | T5 | TF |
| 19 | HW | HW | EM |

To check the effect of these design parameters on the maximum deflection, the natural frequency, and the total weight of the structure, the following three parameters were selected and compared: the design variable with the highest contribution (TD2), one of the contributing variables (D3), and a relatively less-sensitive variable (WF). Then, the parametric analysis was performed. In Figure 8, the variation rates of the structural responses, including the maximum deflection, the total weight, and the first natural frequency, are shown corresponding to a 10% increase of these three parameters, TD2, D3, and WF. In this figure, it is shown that the variation rates of the structural responses concerning these parameters were proportional to the *SRC* results. For example, the changes of the natural frequency, the total weight, and the maximum deflection concerning a 10% change of TD2 were 3.64%, 3.18%, and 8.42%, respectively, while those concerning WF were 0.02%, 0.06%, and 0.04%, respectively. This clearly shows that the *SRC* successfully identifies the rank of the contributing parameters.

*5.2. Reliability Analysis for Target Jacket*

In the reliability analysis, SLS was mainly considered as it was expected to affect the operation of an OWT more frequently than ULS. The tolerance values that determine the serviceability failure can be found in the codes of practice such as DNV and design specifications provided by turbine manufacturers. In this study, a total of five threshold values, i.e., H/200, H/175, H/150, H/125, and H/100, were considered based on the following reference values: H/125 was provided in DNV-OS-J101 code (2014) [32], where H is the projected length of a cantilever beam or the tower height; H/100 was

provided in DNV GL-ST_0126 [4], and the range of H/200–H/125 was provided based on their operating experience [33]. Also, to consider the modeling error, an additional factor was multiplied to the material property, and the partial descriptors were determined based on the relation between the additional random variable and the deflection that was found from a parametric study.

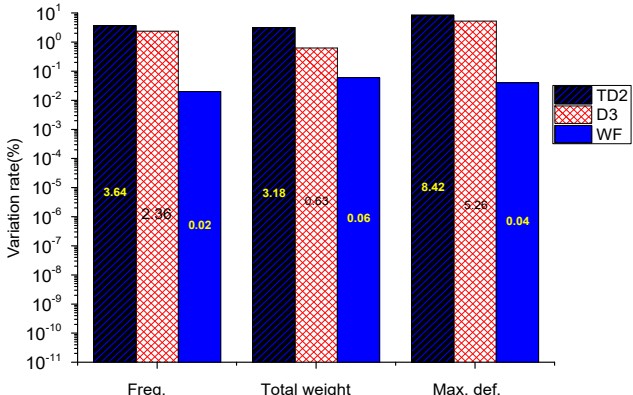

**Figure 8.** The variation rate of the structural responses with respect to the change of design variables.

Table 6 lists the probability results for the five different threshold values for deflection. The number of MCS performed is set to be 5000 considering the time consumption in each FE analysis and the relatively not very low probability values required to be evaluated for SLS compared to ULS.

**Table 6.** Probabilistic results of different thresholds in five cases (5000 MCSs).

| Case | Factors of Ex($\varepsilon$) | Threshold | Failure Probability $P_f$ | Reliability Index $\beta$ |
|---|---|---|---|---|
| 1 | 1 | H/200 | $5.34 \times 10^{-1}$ | $-0.09$ |
| | | H/175 | $2.11 \times 10^{-1}$ | 0.80 |
| | | H/150 | $4.57 \times 10^{-2}$ | 1.69 |
| | | H/125 | $6.25 \times 10^{-3}$ | 2.50 |
| | | H/100 | $3.20 \times 10^{-4}$ | 3.41 |
| 2 | 1.05 | H/200 | $4.10 \times 10^{-1}$ | 0.23 |
| | | H/175 | $1.44 \times 10^{-1}$ | 1.06 |
| | | H/150 | $2.76 \times 10^{-2}$ | 1.92 |
| | | H/125 | $2.46 \times 10^{-3}$ | 2.81 |
| | | H/100 | 0 | >3.5 |
| 3 | 1.1 | H/200 | $3.06 \times 10^{-1}$ | 0.51 |
| | | H/175 | $9.07 \times 10^{-2}$ | 1.34 |
| | | H/150 | $1.26 \times 10^{-2}$ | 2.24 |
| | | H/125 | $8.01 \times 10^{-4}$ | 3.16 |
| | | H/100 | $2.32 \times 10^{-4}$ | 3.50 |
| 4 | 1.15 | H/200 | $2.09 \times 10^{-1}$ | 0.81 |
| | | H/175 | $5.78 \times 10^{-2}$ | 1.57 |
| | | H/150 | $8.45 \times 10^{-3}$ | 2.39 |
| | | H/125 | $5.09 \times 10^{-4}$ | 3.29 |
| | | H/100 | 0 | >3.5 |
| 5 | 1.2 | H/200 | $1.47 \times 10^{-1}$ | 1.05 |
| | | H/175 | $3.33 \times 10^{-2}$ | 1.83 |
| | | H/150 | $4.64 \times 10^{-3}$ | 2.60 |
| | | H/125 | $5.30 \times 10^{-4}$ | 3.27 |
| | | H/100 | 0 | >3.5 |

The analysis results were obtained through performing MCS and the LHS method using the finite element model.

The results provided in Table 5 show that the failure probability increased as the threshold value decreased. It was also observed that the selection of SLS criteria impacted the foundation design and costs. The threshold value should be carefully chosen considering the balance between the operation cost and serviceability.

Table 6 shows that the failure probability decreased when the factor multiplied by the elastic modulus increased. As an example, for the threshold of H/150, the failure probability significantly decreased from $4.57 \times 10^{-2}$ to $4.64 \times 10^{-3}$ when the factor changed from 1 to 1.2.

### 5.3. The Comparison between the Deterministic and Probabilistic Analyses Results

For the jacket substructure described in Section 4, the deterministic analysis results show that the maximum lateral deformation of the tower was 440.1 mm (H/203) in Figure 6, which satisfies the deflection limit H/100 in DNV GL-ST-0126 [4]. However, it is noted that some of the serviceability criteria did not meet the reliability level specified in current international codes. The target reliability index for the serviceability limit state provided in international codes including EN 1990 [34] and ISO 2394 [35] is 1.5, corresponding to a failure probability of $6.6807 \times 10^{-2}$. For example, for Case 1, the calculated failure probability for a threshold of H/200 is $5.34 \times 10^{-1}$, which is considerably more significant than the target failure probability of $6.6807 \times 10^{-2}$. This example shows that the assessment of the foundation design of OWTs should be performed in probabilistic terms. It is also illustrated that the optimized design can be achieved using a careful reliability analysis rather than a simple deterministic check.

### 6. Conclusions

Based on statistical correlation analysis and FE simulations, a reliability analysis framework was proposed to optimize the design for the jacket foundation of OWTs. The statistical correlation analysis was conducted to statistically identify important uncertain parameters affecting the overall stochastic structural performance. Compared to previous probabilistic approaches, the proposed statistical framework did not involve approximations in the FEA or probabilistic analysis, and effectively reduced the dimension of the parameters and better optimized the structural design with respect to the target reliability level.

The FE-simulation was based on a combination of the MCS and the LHS, which does not rely on an approximation of the limit state function or the reliability estimation, and reduced the FEA calls. The proposed reliability analysis framework was applied to a target jacket substructure for a 3 MW OWT. The FE modeling was conducted using a commercial nonlinear FE code ANSYS by scripting in APDL, and the loads applied to the target structure were also provided in this paper. By performing parameter identification, the results showed that the *SRC* measure successfully ranked the contributing parameters. The reliability analysis showed that the selection of SLS criteria impacted the foundation design and costs, and therefore, the threshold value should be carefully chosen considering the balance between the operation cost and serviceability. At last, by comparing the results of deterministic and the probabilistic methods, we could conclude that the assessment of OWT foundation designs needs to be carried out in a probabilistic way. Compared to a simple deterministic check, the uncertainties could be considered in a probabilistic analysis, and thus the optimized design could be achieved in terms of the target reliability level.

**Author Contributions:** J.Z. wrote the manuscript together with W.-H.K., K.S. designed the idea. F.L., provided important materials.

**Funding:** This research was funded by the National Natural Science Foundation of China (grant nos. U1806229, 51678322 and 51779237), the Fundamental Research Funds for the Central Universities and the Taishan Scholars Program of Shandong Province.

**Conflicts of Interest:** The authors declare no conflict of interest.

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
