# Peer review of "Reliability-Based Serviceability Limit State Design of a Jacket Substructure for an Offshore Wind Turbine"

_energies, doi:10.3390/en12142751_

Round 1

Reviewer 1 Report

The manuscript presents a reliability analysis framework for the design of offshore wind turbine jacket structures. The authors combine a statistical correlation analysis to identify relevant/critical design parameters with a Finite Element estimation process. The analysis is carried out in a probabilistic way and the results are compared with a conventional deterministic method. Overall the manuscript is well prepared, however there are some areas that require improvement.

General comments

1.     The authors should enhance the abstract and improve language throughout the manuscript. E.g. Improve English in lines 19-20: “The serviceability….requirements”.

2.     The reference list should be enhanced and include some publications from Energies to highlight relevance to the journal.

3.     It is recommended to use recent references for a field such as offshore wind due to rapid growth. For example, the authors should replace the statistics the first paragraph with recent ones along with reference [1]. A recent global wind report has been published for 2018 from the same body.

4.     Although the growth of the offshore wind sector is important and relevant to the topic covered, fig. 1 does not add value to the manuscript and is not considered necessary. Particularly, when so many statistics are already mentioned in lines 33-40.

5.     Section 2, requires re-structuring as the purpose is not clear. For example, section 2.2 presents some theoretical aspects, which however do not reflect section 2. It is expected that the section should clearly and explicitly describe the proposed framework. Any relevant theoretical aspects can be described within the subsections. Hence, it is suggested to re-name section 2.2.

6.     On the same note, section 2, is entitled “proposed reliability analysis framework” and section 6 is entitled “The proposed reliability-based design framework”. The two sections have the same title and discuss the proposed framework. It is suggested that section 6 is removed and the content (including the flowchart) is incorporated in section 2. This allows for a better structure and presents the framework to the reader early on.

Specific comments

7.     Line 163-164: Why does APDL have an advantage for reliability assessment? Scripting is also available in other commercial FE packages. The authors should justify that, otherwise it is suggested that this line is removed from the manuscript.

8.     Reference [22]. Please ensure that the report is cited correctly in the reference list.

9.     Lines 175-176: Refer to the steel properties. Please add a table after section 3.2 including the steel and soil material properties which were used as input for the FE model.

10.  Line 183: “…..appropriate load simplifications are made”. It is suggested that the authors explicitly state the simplifications made when considering loads. For example, one obvious simplification is that the current is not included. Add a sentence to explain the rationale behind any assumptions/simplifications made.

11.  Line 204: Add the expressions for velocity and acceleration which are required for equation 9.

12.  Line 211: Mention the specific site that you are referring and also add appropriate citation or source.

13.  Line 212: Please add the location of the mass point. I suppose the RNA mass is considered at the top but clarify it in-text.

14.  Line 218: Mention in-text the number of iterations.

15.  Table 1: Correct Unit Mm to mm also present diameter and thickness in separate columns. The diameter does not require ø before the value.

16.  Increase the quality of Fig. 6 and Fig.7. Particularly for Fig. 7 it is a difficult figure to read, so please increase fonts or choose an alternate presentation.

17. It is suggested to include a paragraph in the conclusions, discussing the performance of the presented framework compared to other probabilistic methods. Focus on the advantages and disadvantages of the proposed approach compared to other probabilistic approaches.

Author Response

We thank the editor and reviewers for their time and efforts to review our article. We followed the editor’s instruction to state our item-by-item response to the reviewers’ comments. Our response is provided below, after the reviewer’s comments, and each revision is displayed in red font in the revised version manuscript.

General comments:

1.     The authors should enhance the abstract and improve language throughout the manuscript. E.g. Improve English in lines 19-20: “The serviceability….requirements”.

Response 1: We appreciate the reviewer’s suggestion, and the language of manuscript has been improved, as shown in the revised manuscript.

2.     The reference list should be enhanced and include some publications from Energies to highlight relevance to the journal.

Response 2: We agree with the reviewer’s suggestion that the reference list should be enhanced and include some publications from Energies. The reference list have been updated/improved accordingly, as shown in the reference list of the manuscript.

3.     It is recommended to use recent references for a field such as offshore wind due to rapid growth. For example, the authors should replace the statistics the first paragraph with recent ones along with reference [1]. A recent global wind report has been published for 2018 from the same body.

Response 3: We agree with the reviewer’s comment. We replaced/updated the content based on the recent Global Wind Energy Council annual report. Please refer to page 1 in the revised manuscript.

4.     Although the growth of the offshore wind sector is important and relevant to the topic covered, fig. 1 does not add value to the manuscript and is not considered necessary. Particularly, when so many statistics are already mentioned in lines 33-40.

Response 4: Thanks for the reviewer’s suggestion. We removed the Fig.1, and please refer to page 1 in the revised manuscript.

5.     Section 2, requires re-structuring as the purpose is not clear. For example, section 2.2 presents some theoretical aspects, which however do not reflect section 2. It is expected that the section should clearly and explicitly describe the proposed framework. Any relevant theoretical aspects can be described within the subsections.

Response 5: We agree with the reviewer’s suggestion. The name of former section 2.2 was removed, and the former section 2.2 was combined with section 2.1. Also, the contents have been revised (marked in red texts) and the order of the paragraphs have been rearranged. Please refer to section 2.1.

6.     On the same note, section 2, is entitled “proposed reliability analysis framework” and section 6 is entitled “The proposed reliability-based design framework”. The two sections have the same title and discuss the proposed framework. It is suggested that section 6 is removed and the content (including the flowchart) is incorporated in section 2. This allows for a better structure and presents the framework to the reader early on.

Response 6: Thanks for the reviewer’s suggestion, and the section 6 was combined into section 2. Please refer to page 4.

Specific comments:

7.     Line 163-164: Why does APDL have an advantage for reliability assessment? Scripting is also available in other commercial FE packages. The authors should justify that, otherwise it is suggested that this line is removed from the manuscript.

Response 7: We agree with the reviewer’s comment, and we removed the specified line from the manuscript. Please refer to the first paragraph of section 3.2 in page 6.

8.     Reference [22]. Please ensure that the report is cited correctly in the reference list.

Response 8: The format for reference [22] (reference 25 in the current revised version) has been corrected.

9.     Lines 175-176: Refer to the steel properties. Please add a table after section 3.2 including the steel and soil material properties which were used as input for the FE model.

Response 9: Tables 1 and 2 have been added after section 3.2 to report the soil and steel material properties used in the FE model.

10.  Line 183: “…..appropriate load simplifications are made”. It is suggested that the authors explicitly state the simplifications made when considering loads. For example, one obvious simplification is that the current is not included. Add a sentence to explain the rationale behind any assumptions/simplifications made.

Response 10: In section 3.3., the following sentences have been added: “The distributed loading including wind loading and wave loading is simplified as point loading to deal with the large-scale structural configurations and improve the computational efficiency. In addition, the current loading, which has a minor effect on the jacket foundation compared to wave loading and wind loading, is not considered in this study.”

11.  Line 204: Add the expressions for velocity and acceleration which are required for equation 9.

Response 11: The expression for the velocity of water has been added as Equation (10), and the description of the parameters have also been provided below the equation.

12.  Line 211: Mention the specific site that you are referring and also add appropriate citation or source.

Response 12: The specific site selected in this study is about 5.3 km far from Guishan island (Zhuhai, Guangdong, China), and the project report has been cited.

13.  Line 212: Please add the location of the mass point. I suppose the RNA mass is considered at the top but clarify it in-text.

Response 13The locations of the simplified mass points have been clarified in the text.

14.  Line 218: Mention in-text the number of iterations.

Response 14Seven iterations were running to obtain the required structural parameters. This has been updated in the text.

15.  Table 1: Correct Unit Mm to mm also present diameter and thickness in separate columns. The diameter does not require ø before the value.

Response 15The Unit and structural component sizes have been corrected and reported in Table 3.

16.  Increase the quality of Fig. 6 and Fig.7. Particularly for Fig. 7 it is a difficult figure to read, so please increase fonts or choose an alternate presentation.

Response 16The quality of Fig.7 has been improved by removing the tick labels and re-presenting them with the aid of arrows.

17. It is suggested to include a paragraph in the conclusions, discussing the performance of the presented framework compared to other probabilistic methods. Focus on the advantages and disadvantages of the proposed approach compared to other probabilistic approaches.

Response 17The advantages of the proposed probabilistic approach are discussed in the first paragraph of the conclusions.

Reviewer 2 Report

A reliability analysis to optimize the jacket foundation of off-shore wind turbines (OWT) using statistical correlation analysis and finite element based simulation is very commendable. The results provided in this paper will be of great interest to the readers of this journal. The outcome of the manuscript will help in achieving a proper trade off between the operation cost and serviceability of OWTs with reliability in future OWT design and management. The language handled by the authors are fine and the figures are clear. However, in few of the places, the representations are confusing and justification based on the actual physics are lacking. The reviewer has marked those places in the attached manuscript (marked, see the attachment), In view of the above, the reviewer recommends the manuscript to be considered only after major revision and subsequent review of the revised manuscript. Given the knowledge of the authors on the subject matter, the reviewer believes that the authors could respond at the earliest.

Author Response

We thank the editor and reviewers for their time and efforts to review our article. We followed the editor’s instruction to state our item-by-item response to the reviewers’ comments. Each revised place is displayed in red font in the revised version.

Comment 1: The -14 m curve is not correct. The legend markers are not in red color but in multiple color. It is misleading. Also, it is surprising to see the -14 m to suddenly show a decreasing trend. Some more justification is needed.

Response1: The multiple color mistakes for -14 m curve in Fig. 4 have been corrected in red color. The P-Y curves are determined by the real soil properties reported in Table 1.

Comment 2: Kindly put the abbreviation in the legend also on the caption. It is confusing to read the plot. Also use a consistent abbreviation.

Response2: The abbreviation for design variables have been summarized in Table 4 (Table 2 in the original version).

Comment3: Follow the same suggestion in Fig. 7. Plot the y axis in log scale so that the WF parameter could be visibly available. Log plot in y-axis will be helpful as the parameters are jumping order of magnitudes.

Response3: The variation rate - design variables curves in Fig. 7 are re-plotted in log scale for the y axis.

Round 2

Reviewer 2 Report

The authors have answered the queries of the reviewers to a satisfactory extent. The paper could be considered for publication in the revised format.